# Characterization of Raid Hipico Uruguayo Competencies by Ride Type: Causes of Death and Risk Factors

**DOI:** 10.3390/ani13101602

**Published:** 2023-05-10

**Authors:** Gimena Brito, Juan Pablo Damián, Gonzalo Suárez, Gretel Ruprechter, Pablo Trigo

**Affiliations:** 1Unidad de Análisis Clínicos, Imagenología y Laboratorio de Endocrinología y Metabolismo Animal, Departamento de Clinicas y Hospital Veterinario, Facultad de Veterinaria, Universidad de la República, Montevideo 13000, Uruguay; 2Departamento de Biociencias Veterinarias, Facultad de Veterinaria, Universidad de la República, Montevideo 13000, Uruguay; 3Unidad de Farmacología y Terapéutica, Departamento Hospital y Clínicas Veterinarias, Facultad de Veterinaria, Universidad de la República, Montevideo 13000, Uruguay; 4IGEVET CONICET CC La Plata, Facultad de Ciencias Veterinarias, Universidad Nacional de la Plata, La Plata B1900, Argentina

**Keywords:** horse, fatalities, comfort index, experience, equine, animal welfare

## Abstract

**Simple Summary:**

The aim of this study was to characterize the RHU competencies according to the distance (short vs. long), causes of deaths, and associated risk factors. The studied population comprised 16,856 horses that participated in RHU rides from 2007 to 2018. During the entire period, there were 99 fatalities. The percentage of inexperienced horses and those who completed the ride was greater in short races than in long races. In both types of rides, more horses died during than after the ride, and inexperienced horses were more likely to be dead than horses with prior experience in the sport. Short rides were associated with increased risk of sudden death, while long rides were associated with increased risk of death due to metabolic alterations.

**Abstract:**

RHU is the oldest endurance sport in Uruguay. However, despite 80 years of racing, there are no studies to characterize this type of competition, explore rates and causes of death, and identify the associated risk factors. The aim was to characterize the Raid Hipico Uruguayo (RHU) competencies according to the distance (short (SR, 60 km) vs. long (LR, 80–115 km)), the causes of deaths, and the associated risk factors. The study population comprised horses (*n* = 16,856) that participated in RHU rides from 2007 to 2018. LR were more frequent than SR (*p* < 0.001). The average speed of winners was higher in SR (32.12 km/h) than in LR (28.14 km/h) (*p* < 0.001). There were 99 fatalities (5.9 per 1000 starts). SR had greater frequency of high comfort index (CI = Temp [°F] + Humidity [%]) than LR, and LR had greater frequency of low CI than SR (*p* < 0.001). The percentage of inexperienced horses and those who completed the ride was greater in SR than in LR (*p* < 0.001). In both types of rides, more horses died during than after the ride, and inexperienced horses were more likely to suffer fatalities than horses with prior experience in the sport (*p* < 0.05). SR were associated with increased risk of sudden death, while LR were associated with increased risk of death due to metabolic alterations. The high fatality index shown in this work warrants urgent investigation in this sport to minimize mortality associated with RHU-specific diseases.

## 1. Introduction

Endurance equestrian sports have a long history, but it has experienced great growth in recent decades, mainly FEI endurance. As described by the FEI, “Horsemanship and Horse welfare are the core of endurance riding. Endurance is a test of the Athlete’s ability to manage the Horse safely over an Endurance course. It is designed to test the stamina and fitness of the Athlete and Horse against the track, distance, terrain, climate, and clock, without compromising the welfare of the Horse” [1]. 

Because metabolic disturbances and deaths occur more frequently than in any other type of equestrian discipline, all equine resistance sports have a strict veterinary control that ensures the health of equine competitors [1,2]. A series of veterinary inspections and examinations are established in the interest of the health, safety, and welfare of the horse in the competition. Only competitors whose horses have passed all the inspections and examinations are entitled to be classified in the final list of results [1,3].

Elimination rates appear to have increased over recent years, which is a source of concern for the sport’s ethics and image. Main reasons for elimination are lameness and metabolic disturbances, associated with dehydration and electrolyte disturbances, and with substrate depletion in active muscle fibers. Moreover, there are severe consequences of this metabolic derangement, including heat stroke, rhabdomyolysis, colic, kidney and liver insufficiency, laminitis, and disseminated intravascular coagulation [2,4].

The Raid Hipico Uruguayo (RHU) was the first endurance sport in Uruguay. This sport dates from 1944 and since then, rides have been regulated by the Uruguayan Equestrian Federation (FEU). It is the most popular and typical equestrian discipline of Uruguay, with some unique characteristics. RHU is considered as an endurance discipline. Research into veterinary problems in endurance horses is increasing, but there is almost no information available on endurance races not regulated by FEI. In theory, most of the information generated from FEI endurance races could be applicable to RHU horses. However, many of these practices were not useful in RHU horses, and most management, training and nutrition techniques come from empiric experience without any scientific basis. Something similar occurs with many other equestrian sports that are scientifically overshadowed by disciplines with common characteristics. However, small differences in the sports can produce large differences in metabolic and locomotor behavior, with strong impact on the athletic horse.

Ride distances vary from 60 to 115 km and according to this, FEU has determined 2 categories: short rides (60 km) and long rides (80–115 km), all of which are divided in only 2 phases, being the first phase of long duration (2/3 of the total distance). Winning horses average speeds from 25 to 37 km/h, reaching top speeds of 50 km/h. The winning horse is the first to cross the finish line and meets subsequent veterinary requirements. For the rest of riders, the cut-off time of crossing the finish line is 45 min after the winning horse arrives [3].

The breed of the horses competing in RHU was not officially registered until 2019. In an unpublished study from Brito et al., of 305 horses that raced in 1 season of RHU, 237 were crossbreeds, 39 thoroughbreds, 15 Anglo-Arabians, and 14 Arabians horses. Horses called crossbreeds mostly had more than 75% thoroughbred blood.

The competitions take place on flat terrain with mostly hard surfaces, and the minimum weight of the riders should be 85 kg. Horses and riders can compete in any ride, regardless of their previous experience. Horses are examined by official veterinarians before the ride, after the first phase, and at the end. Veterinary control after the first phase is performed 20 min after the arrival of the horse, and once passed, horses must wait a 40-min compulsory rest period before starting the last phase. Horses are eliminated from the ride if veterinarians consider their metabolic status or orthopedic condition not to be adequate to enable them to continue the ride [3]. Sixty-one percent of the participants do not finish the ride due to lameness or metabolic reasons [5], which can sometimes lead to fatalities.

From the point of view of health and welfare, the death of equines during or after competition is a major concern for vets, riders, organizers, and for spectators. Studies on the causes of death in sports horses are scarce and mostly refer to racehorses, being the main causes: sudden death [6,7] and catastrophic musculoskeletal injury [8,9].

Fatalities during endurance exercise are recognized as a consequence of prolonged exercise, but data documenting incidence and causes are very limited. Balch et al. (2019) reported 127 fatalities out of 335,456 starts (0.28 fatalities per 1000 starts) during the period 2002 to 2018 [10]. According to Balch et al. (2019), 77% of deaths were attributed to the high demands of endurance exercise (leading to severe muscle cramping and exhaustion, mostly attributable to acute abdominal pain) and 33% due to injuries not associated with the metabolic demands of endurance exercise (such as falling off a cliff or the trail, kick injury) [10]. In addition, the risk of death increased with the distance traveled (0.12, 0.35, and 1.48 fatalities per 1000 starts in rides of 48, 80, and 160 km, respectively) [10].

Although this sport (RHU) has been carried out in Uruguay for several decades, there are no reports characterizing the ride conditions, as well as the causes of death and their risk factors. We hypothesize that although this sport has similarities with other endurance disciplines, it has very different characteristics that affect the causes of death during races. Therefore, the aim of this study was to characterize the RHU competencies according to the distance (SR vs. LR), the causes of deaths, and the associated risk factors.

## 2. Materials and Methods

The study was performed with the endorsement of the FEU, guaranteeing confidentiality regarding the names of the horses and owners.

### 2.1. Data Collection and Studied Variables

This was a retrospective cohort study. Data from all RHU rides from 2007 to 2018 were collected from FEU archives. All rides were contested annually between the months of March to December. The information obtained was entered into a computerized database, and each RHU ride was assigned a unique identification number. The database included the temperature and humidity, ride type and length, number of horses that started, number of eliminated and retired horses (total, by phases and reason), number of inexperienced horses (no RHU racing experience), number of horses that completed the ride and average speed (km/h) of each phase (phases 1 and 2), and average speed of the winning horse. If a variable was not recorded, the variable was assigned a value of not available. Reports with incomplete data for more than two variables were not included in the study. When fatalities occurred, the cause of death was recorded. To establish the cause of death, necropsy examinations by official veterinarians were performed on all dead equines. All data were obtained from FEU reports.

### 2.2. Variable Categorization

Comfort index (CI) was calculated by the sum of the temperature in Fahrenheit degrees and the relative humidity as a percentage [11]. CI was classified into three categories: low (CI < 130), medium (CI 130–150), and high (CI > 150). Ride types were classified by the length into two categories according to the FEU designation: short ride (SR: 60 km) and long ride (LR: 80–115 km). Causes of death were classified in four categories [9]: metabolic conditions (colic, exhausted horse syndrome, disseminated intravascular coagulation), catastrophic musculoskeletal injuries (defined as horses that died or were euthanized due to severe acute bone fractures that carry a poor clinical prognosis), sudden death (defined as acute death in a closely observed and previously apparently healthy animal), and accidental.

Additionally, the part of the race where the death occurred was classified as during the ride (phase 1 or phase 2) or after the ride (24 h after the finish of the ride) to know if the horses that suffered fatalities completed the course or not.

### 2.3. Statistical Analysis

Descriptive analyses (mean and standard deviations or percentage values) were calculated for the variables CI, horse experience, completed ride, speed (phase 1, phase 2, and average), overall fatalities, cause of death, and ride type. Statistical differences by ride type were calculated using a Chi-squared test, Fisher’s test, or Wilcoxon rank sum test. Screening of all exposure variables for overall fatalities (Live/Death) were performed separately using univariable Logistic Mixed Model analysis. Only variables with *p* < 0.2 were considered for inclusion in the multivariate Logistic Mixed Model analysis. In both analyses, all variables were considered fixed effects and the ride was considered as a random effect. The multivariate models were built using a forward selection procedure whereby variables with a Wald-test *p* < 0.05 were retained in the model. *p*-values of less than 0.05 were considered statistically significant. All analyses were performed in R (Version 4.2.2, 2022) and RStudio (version 2022.12.0 Build 353) [12].

## 3. Results

From a total of 702 RHU competitions taking place between 2007 and 2018, there was a 3-fold greater frequency (*p* < 0.0001) of LR (509, 42 rides per year) than SR (193, 16 rides per year). There were 16,856 horse starts, of which the number of horses was also greater in LR than in SR (Table 1, *p* < 0.0001). The average of horses competing per ride was 22 in SR and 29 in LR. Overall, 40.5% of the horses completed the ride, 43.1% were not able to complete the course due to metabolic reasons, 12.3% did not complete because of lameness, while 4.1% of the horses were retired from the ride due to rider-decision. The average speed of the winning horses was 28.1 km/h and 32.1 km/h (Table 1), with maximum and minimum speeds of 32.6 km/h and 20.3 km/h for LR and 35.9 km/h and 25.5 km/h for SR. The highest average speed in each phase and the average speed of the winning horse were recorded in SR (Table 1).

Over the 12-year study period, there were 99 fatalities, and 68 of these horses were euthanized. All the euthanasias were performed with a rigorous criterion evaluating the future life and the suffering of the horse. The risk of fatality over the entire period was 5.87 per 1000 starts. The average number of deaths per year were 8.25 and did not differ over the years studied. The risk of fatality was significantly greater (*p* = 0.05, odd ratio = 1.52) for participation in SR (7.9 fatalities per 1000 start) than in LR (5.2 fatalities per 1000 start) (Table 2). There were significant differences in causes of death by ride type (Table 1). Short rides were associated with a greater proportion of sudden death, and LR were associated with a greater proportion of deaths due to metabolic alterations (Table 1). Catastrophic injuries occurred in a high proportion in both ride types (Table 1). Of the total deaths for metabolic reasons, 24 (69%) of the fatalities developed acute abdominal pain, 7 (20%) equine exhausted syndrome, and 4 (11%) disseminated intravascular coagulation.

CI varied with the type of ride, SR had a greater frequency of high CI in comparison to LR; and LR had a greater frequency of low CI than SR, while there were no differences between ride type for the medium CI (Table 1). However, when CI was evaluated separately for each ride type, CI did not represent a significant risk factor for death in either ride type (Table 2).

Inexperienced horses were more likely to suffer fatalities than experienced horses in both ride types (Table 2 and Table 3).

There was no association between experience and the probability of dying in LR (Table 2), but under a multivariate analysis (Experience and Completed ride), a significant association was found between these variables in LR (Table 3).

Horses that participated in SR completed the ride in a greater proportion than those that participated in LR (Table 1). Besides, regardless of ride type, more horses died during the ride than after them, so most of them did not complete the ride (Table 2 and Table 3).

## 4. Discussion

This is the first study to characterize RHU rides according to ride-type (short vs. long), as well as the causes of death and their risk factors. This was a retrospective cohort study, where many variables, such as sample population and variability associated with horse background and the environment of the races, could not be recorded. The extent of the limitation should be considered for the interpretation of the results.

Equestrian endurance sports require the greatest physiological demand for the athlete horse FEI [2,4,13,14]. The FEI endurance is the most widely described [2,4,15,16,17]. The racing distances were classified as short and long according to the FEU regulations. Long FEI endurance races would be longer than 140 km. Compared to FEI endurance races, RHU has higher speeds, greater weight load, fewer stages, a lower proportion of horses that finish the race, and a higher fatality rate. According to our data, RHU is most likely one of the most demanding events for horses [5].

During the period 2007–2018, LR were more frequent than SR, which follows the same pattern as FEI endurance 14. However, unlike endurance, where the speeds of the races were below 25 km/h [14,18,19,20], RHU races were faster, with speed averages between 28 km/h and 32 km/h, reaching maximum speeds close to 36 km/h. In addition, horses completing RHU races were 10 to 40% less than those reported for endurance [20,21,22]. Another clear difference with endurance is that while in RHU the main cause of elimination was due to metabolic condition (43%), in endurance the highest percentage of elimination (25 to 40%) was due to lameness [14,21,22,23].

Among the main RHU characteristics, the following are briefly highlighted: LR are more frequent than SR; SR are faster than LR ones; SR had a higher frequency of high CI than LR, and LR had a higher frequency of low CI than SR; and the percentage of inexperienced horses and those who completed the ride was greater in SR than in LR.

Regarding fatalities, there were 99 deaths, of which 68 were euthanasia. In both types of rides, more horses died during than after the ride. Most causes of fatalities are incompatible with the endurance exercise. The horses that suffered fatalities and finished the course were due to metabolic causes, except for one animal that presented sudden death. The higher rate of occurrence of metabolic fatalities in LR most likely induced the higher course completion in horses that died during LR (Table 1 and Table 2).

The probability of suffering fatalities was higher in inexperienced horses. In addition, SR were associated with increased risk of sudden death, while LR were associated with increased risk of death due to metabolic alterations. It is interesting to note that although equine resistance sports are very popular throughout the world and that their popularity has been growing [24], according to our knowledge, except for one study [10], there are no reports on equine fatalities in endurance equestrian sports. During the 12-year study period, the fatality rate in RHU was 5.87 per 1000 starts.

This result is much greater than that reported by Balch et al. (2019) in endurance horses under AERC rules (0.28 fatalities per 1000 starts) [10]. Horse deaths in RHU competitions attract the attention of public opinion and negatively affect the sensitivity of the public towards these sports; they also generate controversial opinions regarding the intensity of the exercise carried out by the animals that participate.

The higher fatality rate of the RHU compared to the endurance competition once again highlights the high metabolic and locomotor demand that this sport represents for the horse. The metabolic requirements demanded by high speeds over long distances, with only two stages, create highly challenging conditions for the RHU horse. Additionally, horses run mostly on hard ground, with a high rider weight. Another characteristic of the sport is the important prize money, which adds excitement to the already exciting competition, and can distract competitors from the horse’s state of health and well-being. Since the death of equines during competitions represents a major welfare concern, it should be a priority to know the frequencies and causes of death in all equine sports worldwide. It is a great challenge for veterinarians to try to minimize the frequency of equine deaths during sports. In this sense, the fatalities in the RHU show the need for greater controls and strictness in the limits to which equines are exposed.

In this study, the type of the races (SR vs. LR) in RHU was influenced by the comfort index, the percentage of inexperienced horses, whether they completed the race, the speeds, and the number and cause of fatalities. The SR had higher speed, higher frequency of high comfort index, greater percentage of inexperienced horses, and those who completed the ride in comparison with LR. The comfort index was used as an indicator of thermal stress in this work. It is widely used due to its simplicity and low cost, but there are significant weaknesses due to misleading for many combinations of temperature and humidity. Many indices have been evaluated to assess heat stress, even in horses, showing a better predictive capacity. This limitation should be considered for the interpretation of the results.

The risk of death of horses was higher in SR than in LR, unlike what was reported in endurance by Balch et al. (2019), in which the risk of death increased with the distance traveled (0.12, 0.35, and 1.48 fatalities per 1000 stars in rides of 48, 80, and 160 km, respectively) [10]. It is possible to speculate that, although less distance is covered, the big locomotor and metabolic demand imposed by a greater speed in SR than in LR determined a greater proportion of equine deaths—mainly catastrophic and sudden deaths. According to the type of death, in relation to the type of race, SR had some similarities with racehorses, since deaths due to catastrophic injuries and sudden death predominate [2]. On the other hand, LR agreed with what has been reported for endurance events where the highest proportion of causes of death were metabolic alterations and catastrophic injuries [6,7]. In previous studies [24,25] no associations were found between the speed of individual horses and elimination for lameness or metabolic reasons. Although in our study, the speed of the races in which horses died did not differ statistically from those in which they survived, we cannot conclude that speed does not influence the deaths of horses. Recorded speed was the average speed of the stage in which they died, or the average speed of the race if they managed to finish it. Therefore, the individual speed of the horse was not considered, nor was the accumulated distance—which requires another study design for its analysis.

The highest proportion of deaths in SR was due to sudden death, defined as acute, exercise-associated death in a closely observed and previously apparently healthy animal [26]. Cardiovascular disease is often confirmed or suspected [27]. Most of the studies on sudden death are in thoroughbred racehorses and eventing horses [7,28]. According to Lyle et al. (2011), the prevalence of sudden death in racehorses in the UK between 2000 and 2007 was 0.28 deaths per 1000 starts [6]. Comyn et al. (2017), found a prevalence of 0.14 death per 1000 starts in FEI eventing horses between 2008 and 2014 [29]. In the present study, the mortality rate from this cause was 1.02 deaths per 1000 starts, which is higher than any previous report. Intense exercise requires high oxygen consumption such as horse racing or three-day events, producing large increases in cardiac output and blood pressure, increasing propensity for major cardiovascular events [6,29].

Navas de Solis et al. (2018) studied cases of sudden death in many types of equestrian sports and riding horses, and reported that 71.9% of the horses that suffered sudden death collapsed during exercise [27]. These results are in contrast to the work of Lyle et al. (2011) on racehorses, where most sudden deaths occurred after the race [6]. Exercise time during competitions in racehorses is much less than in three-day eventing or RHU. The intensity and duration of the exercise most likely play an important role in the incidence and moment of presentation of sudden deaths in sport horses. Horses collapsing during exercise may suffer catastrophic lesions, and riders may fall and be injured during these episodes and while riding a horse that presented sudden death. Therefore, the study of the causes of sudden death during RHU, and its prevention may be imperative, not only for increasing the horse’s welfare, but also for human safety concerns.

The most frequent cause of death in LR was metabolic condition. This cause of death occurs mainly in horses after long duration submaximal exercises [2,4]. The marked increase in metabolism for such a long period of time is accompanied by intense energy consumption and loss of body water and electrolyte stores, as a consequence of thermoregulation [2,4,10,14,15]. Mild dehydration is compatible with competitive performance, however, serious medical problems may develop in horses that compete in endurance events—even death [2,15]. The distance, due to the longer time in exercise, most likely has an influence on the presentation of metabolic alterations, since it represents a risk factor for its presentation in endurance horses [14]. In our work, the proportion of deaths due to metabolic alterations was higher in LR.

The main cause of death due to metabolic disorders was Acute Abdomen Syndrome (AAS). Balch (2019) reported AAS as being the most common clinical presentation (85%) as a prelude to death in AERC endurance races [10]. The management of the horse before and during the race can lead to episodes of AAS, such as gastric dilatation due to the effect of fasting before the race and peristalsis problems due to dehydration and electrolyte imbalances [2,15].

Catastrophic injuries occurred in a high proportion in both ride types. In this study, the animals with catastrophic injuries did not complete the ride (43/44), being euthanized when an irreparable injury was diagnosed. Of the total catastrophic injuries, 11 were in SR and 33 in LR, although no differences were found. In the present work, the mortality rate due to catastrophic injuries was 2.6 dead horses per 1000 starters, which is even higher than those reported in racehorses or in three-day eventing horses.

According to Misheff et al. (2010), the risk of suffering from bone pathologies during endurance races increases with the distance covered and the increasing speeds [30]. Most epidemiological studies show a positive association between the risk of catastrophic injuries and distance covered, although not all findings have been consistent [31].

In our study, we also show that experience is a risk factor for fatalities. Therefore, another element that could have contributed to the higher risk of death in SR was the greater proportion of inexperienced horses in comparison to LR. Previous studies in sporting horses as endurance [25] or thoroughbred racehorses [32,33] have found experienced horses to be associated with reduced odds of deleterious outcomes compared to horses that were inexperienced (in endurance for the distance) or less experienced thoroughbred racehorses. However, the concept of inexperience is not the same, since in FEI races there is a qualification system that does not allow horses to compete if they have not completed shorter distance races. Therefore, they are only inexperienced at that distance. In our case, although many animals had competed in other sports, none of the horses considered inexperienced had competed in RHU races. The fatality rate for inexperienced horses was 13.2 and 8 fatalities per 1000 starts in SR and LR, respectively.

Based on this set of results, it is clear that although LR are more frequent than SR in RHU, the risk of death is higher in SR than in LR. Therefore, the need to control racing conditions is evident, especially in SR, where speeds are higher.

The differences and characteristics of the RHU as an equestrian endurance sport shown in this study, with notable consequences for equine athletes, warn about the importance of independent evaluation of each discipline to ensure the well-being of our equine and human athletes in all equestrian sports. Further studies are needed to assess more specific risk factors, in order to provide objective data that can help to plan racing schedules and serve as a basis for regulations, ultimately improving both the welfare of RHU horses and the public perception of this discipline.

## 5. Conclusions

SR and LR have important differences that are manifested in the characteristics of the race, the causes of death, and the risk of fatality. This must be considered to understand the physical impact of equine participation in this type of event.

The high fatality index shown in this work, especially in inexperienced horses, warrants urgent investigation in this sport to minimize the mortality associated with RHU-specific diseases and to improve the welfare of the RHU horses.

## Figures and Tables

**Table 1 animals-13-01602-t001:** Characteristics of competition by ride type in over 702 Raid Hipico Uruguayo (RHU) rides between competition seasons 2007 and 2018 in Uruguay.

		Competition by Ride Type		
Variable	N	Short (60 km) N = 4030	Long (80–115 km) N = 12,826	*p*-Value ^1^
Comfort Index, *n* (%)	15,628			<0.001
Low		842 (23%) ^A^	3274 (27%) ^B^	
Medium		1683 (46%)	5407 (45%)	
High		1134 (31%) ^A^	3288 (27%) ^B^	
Unknown		371	857	
Experience, *n* (%)	14,124			<0.001
Inexperienced		1754 (52%)	3511 (33%)	
Experienced		1641 (48%)	7218 (67%)	
Unknown		635	2097	
Completed ride, *n* (%)	16,856			<0.001
Yes		1820 (45%)	5001 (39%)	
No		2210 (55%)	7825 (61%)	
Speed Phase 1, (kmhr^−1^) *	16,816	31.97 1.49)	28.24 (1.32)	<0.001
Speed Phase 2, (kmhr^−1^) *	16,816	32.7 (3.1)	28.1 (2.4)	<0.001
Average Speed, (kmhr^−1^) *	16,856	32.12 (1.61)	28.14 (1.38)	<0.001
Overall fatalities, *n* (%)	16,856			0.049
Live		3998 (99%)	12,759 (99%)	
Death		32 (0.8%)	67 (0.5%)	
Causes of Death, *n* (%)	99			<0.001
Accidental		2 (6.2%)	1 (1.5%)	
Catastrophic		11 (34%)	33 (49%)	
Metabolic		5 (16%) ^A^	30 (45%) ^B^	
Sudden		14 (44%) ^A^	3 (4.5%) ^B^	

^1^ Pearson’s Chi-squared test; Wilcoxon rank sum test; Fisher’s exact test. * Speed is expressed in mean (SD). Different uppercase letters within a row indicate significant differences (*p* < 0.05).

**Table 2 animals-13-01602-t002:** Characteristics of overall fatalities by ride type in over 702 Raid Hipico Uruguayo (RHU) rides between competition seasons 2007 and 2018 in Uruguay.

	Short Race			Long Race		
Variable	Live, N = 3998	Death, N = 32	*p*-Value ^1^	Live, N = 12,759	Death, N = 67	*p*-Value ^1^
Comfort Index, *n* (%)			0.60			0.20
Low	834 (23%)	8 (26%)		3261 (27%)	13 (21%)	
Medium	1667 (46%)	16 (52%)		5381 (45%)	26 (42%)	
High	1127 (31%)	7 (23%)		3265 (27%)	23 (37%)	
Unknown	370	1		852	5	
Experience, *n* (%)			0.022			0.11
Inexperienced	1731 (51%)	23 (72%)		3483 (33%)	28 (42%)	
Experienced	1632 (49%)	9 (28%)		7179 (67%)	39 (58%)	
Unknown	635	0		2097	0	
Completed ride, *n* (%)			<0.001			<0.001
Yes	1819(45%)	1 (3.1%)		4995 (39%)	6 (9.0%)	
No	2179 (55%)	31 (96.9%)		7764 (61%)	61 (91%)	

^1^ Pearson’s Chi-squared test; Fisher’s exact test; Wilcoxon rank sum test.

**Table 3 animals-13-01602-t003:** Multivariate mixed logistic regression model of associations between overall fatalities (live/death) by ride type and predictor variables (competition and completed ride) in 702 Raid Hipico Uruguayo rides between competition seasons 2007 and 2018 in Uruguay.

	Short Race			Long Race		
Predictors	Odds Ratios	CI	*p*	Odds Ratios	CI	*p*
(Intercept)	0.00	0.00–0.00	<0.001	0.00	0.00–0.00	<0.001
Inexperienced (YES)	22.90	6.95–75.43	<0.001	37.72	18.93–75.16	<0.001
Completed ride (YES)	0.01	0.00–0.06	<0.001	0.00	0.00–0.00	<0.001
Random Effects						
N Ride		166			447	
Observations		3395			10,729	
Marginal R2/Conditional R2		0.076/0.934			0.034/0.969	

## Data Availability

Data used in this work are property of the Federacion Ecuestre Uruguaya.

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
