# Peer review of "Characterization of Raid Hipico Uruguayo Competencies by Ride Type: Causes of Death and Risk Factors"

_animals, 2023, doi:10.3390/ani13101602_

Round 1

Reviewer 1 Report

The authors have addressed the important extremely important topic of the risk of injury and death of horses during endurance events. There is a lack of such comprehensive studies in the available literature, so the article is an interesting source of information and can be an inspiration for other researchers dealing with the welfare of sport horses. The results presented are also of application value and can be used to effectively manage risk factors for equine death during sport competitions.

The title indicates the aim of the manuscript, the objective is clearly formulated. The methodology is correct and the results are described comprehensively. Conclusions are consistent with the evidence presented. Bibliography is sufficient.

Author Response

The only changes we have made to the manuscript are at the request of the reviewers, but with 4 reviewers, there are many changes throughout the entire document.

We sent the manuscript to be reviewed by a native English speaker. We have not yet received the return of these corrections, the manuscript will be ready as soon as possible.

Thank you very much for the work on reviewing the manuscript. 

Your comments are appreciated as well.

Reviewer 2 Report

While the study is interesting and easy to follow, as a reader at this point I would find minimal, if any application to the majority of the equine industry. The authors fell short in reporting the relevancy of this study, other than to record incidences within one specific area of the sport horse industry only found within one country. Even from a researcher perspective, it is unclear as to where to build from this study as indications concerning future directions are lacking within the current manuscript. The authors are challenged to find ways to demonstrate relevance to a wider spectrum of the equine industry and relevance to researchers looking to build upon this work. 

Reimaging the title would be a starting point in meeting this challenge as most readers would overlook this study unless they were familiar with this particular sport and associated with the country of origin. In addition, the dates are unnecessary within the title. Along with the title missing the ability to draw the reader into the value of this work, the introduction lacks true justification for the study. Other than it being a long standing sport within one country, why is this study necessary? Have there been documented issues within the sport and would this study have an impact on other sports? Why after 80 years of this competition would it now be time for this study? The study concerning endurance horses is mentioned, but how did that study impact the industry and is that the hopes of the current authors with this study. Are there other impact studies concerning other sports that have resulted in improvement within their respective sports and how these sports are conducted and how the animals are trained and/or cared for? Welfare is brought up within this manuscript, and thus, how will this study help improve upon welfare and was this a prior concern within this industry? Additional references concerning other sports impacted by similar studies would be useful beyond just the endurance study. Furthermore, a hypothesis statement should be added at the end of the introduction.

As for the materials and methods, while the authors state "FEU" is "guaranteeing confidentiality", what does that entail? Were participants made aware that they and their horses were a part of this study and how were their identities kept confidential? Was an Institutional Review Board concerning research consisting of human subjects involved in assessing the research protocol prior to the study to ensure ethical use of human subjects and their data was carried out? Were participants given a consent form concerning the use of their data? 

Further details need to be given concerning the races utilized for this study. Include locations, specific dates, and entry limits concerning numbers and other specifics that might limit participation of an individual and/or a horse. A table would be useful for giving additional details concerning which races were included within this study. Were the RHU rides selected from every race offered between 2007 to 2018 or were certain races not included within the study? When was this data collected, ie. during or after the ending date of the races utilized, and how was this data collected? Is it publicly available or did FEU provide this information? What were the steps in requesting, receiving, and assessing the data? What groups were worked with in retrieving data and was it the same group/individuals for all the years studied? 

For the necropsy examinations, was this done prior to the study? If so, clearly state that. Authors state "To establish the cause of death", but was that for research purposes, for the purposes of FEU, or for the owner's purpose? If it wasn't specifically for this study, then, it is hard to note specifically the reason for the necropsy. If the necropsies reported within this study, were specifically for this study, then, documentation of an institutional review of ethical use of animals for research purposes needs to be given, similar to the statement concerning the use of human subjects. However, it would be the assumption of this reviewer that these necropsies were done prior to this study. Thus, in that case, the authors are looking at multiple veterinarians performing these examinations in which this would result in a wide range of inconsistencies between necropsy procedures and techniques. How was this minimized and controlled for to ensure results of these necropsies weren't subject to variability associated with the inconsistencies between veterinarians' procedures, techniques, and experiences? Similarly, the intent of the owners would play a role in how extensive of a necropsy examination was done if these necropsies were not financed and mandated by the race, the study, and/or FEU. In line 145 "rigorous criterion" is mentioned, but not further explained. How was this ensured and what did it entail? Furthermore, in line 110 concerning sudden deaths the authors mention "previously apparently healthy animals", but how was that objectively measured? How was "previously" achieved in that were veterinary records prior to the death acquired? How was "healthy" defined, and was that based off of the necropsy? Was that definition the same for all veterinarians performing the necropsy? 

As for results, rephrase lines 135-136 as unsure how "without commitment to the horse's health or welfare" was determined. For table 1, use the same font type throughout. With percentages given within parentheses for table 1 for most of the data, it is confusing have something different for speed. Assuming the numbers within the parentheses is the standard deviation for speed. May want to move speed data to a separate table. Also, within methods define how objective measures for speed was performed. Note that at the bottom of table 1 it says that "uppercase letters" designates significance, but lowercase letters were given within the table. Finally, within the results, lines 165-166, a title was given for table 1 again, but it was given after table 2 and before table 3, thus, remove.

As for the discussion, in line 181 add in brackets for references 9 and 10. Lines 193-200 repeat the results without giving additional interpretation. Either combine the results with the discussion or introduce within the discussion further interpretation of the data as it relates to other sports and as it relates to the welfare of these animals. The discussion section lacks relevancy to the readers and to a wider audience within the equine industry. While comparison with the endurance industry is given, it is limited in the discussion and does not address further studies reporting sports that may have fatalities, particularly those where speed is a component of the sport. It also lacks interpretation as it relates to utilizing this data for future studies. Lines 210-211 mention "the highly demanding degree that this sport represents for the horses", but what aspect of the sport are you referring to, ie. the speed, environment, etc? If speed is the "demanding degree", for example, how does that relate to other studies evaluating racehorses and fatalities and/or non-completion? The discussion spent much of the time repeating the results without further interpretation and comparisons to other research. The discussion should also include addressing study limitations. Being that this appears to be data taken after the fact, then, variability on what was documented, who documented it, and what information was given and recorded can impact the results. Also, since the study was over an expansive period, while it increased numbers, over the years methods within the race or within recording data could change impacting potential results more so than what was actually tracked and reported. Finally, no pre- collection was done to determine prior health nor were the animals tracked for an exceptionally long period to determine impact. In fact, it wasn't very clear how long after the race animals were tracked for those fatalities reported concerning after race deaths. Without a controlled study it is hard to contain potential variability that needs to be accounted for. Nevertheless, the study can still have merit concerning impact of the industry, even outside of this specific type of race, and impact for future studies, and thus, the authors need to clearly state these impacts within the discussion and summarize within the conclusion statement. 

Author Response

We acknowledge the reviewer’s criticism and we state all corrections and comments below. We are now sending the revised version that we hope fulfill the requirements of ANIMALS. In order to be clearer, we have left all correspondence during this revision, and in italics letters our response.

The only changes we have made to the manuscript are at the request of the reviewers, but with 4 reviewers, there are many changes throughout the entire document.

We sent the manuscript to be reviewed by a native English speaker. We have not yet received the return of these corrections, the manuscript will be ready as soon as possible.

While the study is interesting and easy to follow, as a reader at this point I would find minimal, if any application to the majority of the equine industry. The authors fell short in reporting the relevancy of this study, other than to record incidences within one specific area of the sport horse industry only found within one country. Even from a researcher perspective, it is unclear as to where to build from this study as indications concerning future directions are lacking within the current manuscript. The authors are challenged to find ways to demonstrate relevance to a wider spectrum of the equine industry and relevance to researchers looking to build upon this work. 

Reimaging the title would be a starting point in meeting this challenge as most readers would overlook this study unless they were familiar with this particular sport and associated with the country of origin. In addition, the dates are unnecessary within the title. 

Along with the title missing the ability to draw the reader into the value of this work, the introduction lacks true justification for the study. Other than it being a long standing sport within one country, why is this study necessary? Have there been documented issues within the sport and would this study have an impact on other sports? Why after 80 years of this competition would it now be time for this study? The study concerning endurance horses is mentioned, but how did that study impact the industry and is that the hopes of the current authors with this study. Are there other impact studies concerning other sports that have resulted in improvement within their respective sports and how these sports are conducted and how the animals are trained and/or cared for? Welfare is brought up within this manuscript, and thus, how will this study help improve upon welfare and was this a prior concern within this industry? Additional references concerning other sports impacted by similar studies would be useful beyond just the endurance study. Furthermore, a hypothesis statement should be added at the end of the introduction.

We regret that the reviewer did not find relevance to our work, and consequently we have worked on its relevance.We remove dates from the title

The following sentences were added to the introduction

It is considered as an endurance discipline and therefore both the scientific knowledge and the experience of FEI endurance racing strongly influence the regulations, veterinary management, training and feeding of the horses that compete in RHU. Something similar occurs with many other equestrian sports that are conceptually absorbed by disciplines with common characteristics. However, small differences in sport can produce large metabolic and locomotor differences, which, if not taken into account, have a strong impact on the athlete horse

We hypothesize that although this sport has similarities with other endurance disciplines, it has very different characteristics that affect the causes of death during races.

The differences and characteristics of the RHU as an equestrian endurance sport shown in this study, with notable consequences for equine athletes, warn about the importance of independent evaluation of each discipline to ensure the well-being of our equine and human athletes in all equestrian sports. Further studies are needed to assess more specific risk factors, in order to provide objective data that can help planning racing schedules and serve as a basis for regulations, ultimately improving both the welfare of RHU horses and the public perception of this discipline.

As for the materials and methods, while the authors state "FEU" is "guaranteeing confidentiality", what does that entail? Were participants made aware that they and their horses were a part of this study and how were their identities kept confidential? Was an Institutional Review Board concerning research consisting of human subjects involved in assessing the research protocol prior to the study to ensure ethical use of human subjects and their data was carried out? Were participants given a consent form concerning the use of their data? 

This work is part of a doctoral thesis, and for this particular trial, the University of the Republic did not request authorization from the ethics committee. I submitted an approval request to the institutional committee three days ago to get this sorted out. It will be ready, if approved, in 15 days approx.

It is a retrospective cohort study. We requested the information from the FEU (the signed authorization from the FEU for the use of the data was sent). The information was provided without the names of the horses or the owners (only numbers). The owners and managers of the dead horses signed an informed consent on the data of their animals along with the authorization for necropsy and euthanasia.

Further details need to be given concerning the races utilized for this study. Include locations, specific dates, and entry limits concerning numbers and other specifics that might limit participation of an individual and/or a horse. A table would be useful for giving additional details concerning which races were included within this study.

All horses that competed at RHU between 2007 and 2018 were entered. The following paragraph was entered at M&M

This was a retrospective cohort study. Data from all RHU rides from 2007 to 2018 were collected from FEU archives. 

Were the RHU rides selected from every race offered between 2007 to 2018 or were certain races not included within the study? When was this data collected, ie. during or after the ending date of the races utilized, and how was this data collected? Is it publicly available or did FEU provide this information? What were the steps in requesting, receiving, and assessing the data? What groups were worked with in retrieving data and was it the same group/individuals for all the years studied? 

All horses that competed at RHU between 2007 and 2018 were entered. During the period used, there were no regulatory changes, and the data collection by the FEU was similar. The official work teams (veterinarians, judges and stewards) underwent subtle and gradual changes.

For the necropsy examinations, was this done prior to the study? If so, clearly state that. Authors state "To establish the cause of death", but was that for research purposes, for the purposes of FEU, or for the owner's purpose? If it wasn't specifically for this study, then, it is hard to note specifically the reason for the necropsy. If the necropsies reported within this study were specifically for this study, then, documentation of an institutional review of ethical use of animals for research purposes needs to be given, similar to the statement concerning the use of human subjects. However, it would be the assumption of this reviewer that these necropsies were done prior to this study. Thus, in that case, the authors are looking at multiple veterinarians performing these examinations in which this would result in a wide range of inconsistencies between necropsy procedures and techniques. How was this minimized and controlled for to ensure results of these necropsies weren't subject to variability associated with the inconsistencies between veterinarians' procedures, techniques, and experiences? Similarly, the intent of the owners would play a role in how extensive of a necropsy examination was done if these necropsies were not financed and mandated by the race, the study, and/or FEU. In line 145 "rigorous criterion" is mentioned, but not further explained. How was this ensured and what did it entail? Furthermore, in line 110 concerning sudden deaths the authors mention "previously apparently healthy animals", but how was that objectively measured? How was "previously" achieved in that were veterinary records prior to the death acquired? How was "healthy" defined, and was that based off of the necropsy? Was that definition the same for all veterinarians performing the necropsy? 

Necropsies were mandatory for all dead horses. A signed consent authorization for the use of the data and potential sampling was requested. The FEU veterinarians are all equine specialists with experience in field work, however their expertise was not controlled. Several veterinarians performed these necropsies. Every year an annual meeting is held to agree on criteria and protocols for necropsy, euthanasia and diagnosis and treatment. The necropsy data are for information only, they were not submitted to analysis or discussion.

Health within the event is regularly controlled within the regulatory conditions. A healthy horse in the race is one that is in the competition (from the initial inspection until 24 hours after the end) and has satisfactorily passed the mandatory controls (initial - phase control - final inspection) or is under veterinary control. either by decision of the rider or a judge. In addition, riders must report any suspicion of compromise to the health of their animals.

As for results, rephrase lines 135-136 as unsure how "without commitment to the horse's health or welfare" was determined. For table 1, use the same font type throughout. With percentages given within parentheses for table 1 for most of the data, it is confusing have something different for speed. Assuming the numbers within the parentheses is the standard deviation for speed. May want to move speed data to a separate table. Also, within methods define how objective measures for speed was performed. Note that at the bottom of table 1 it says that "uppercase letters" designates significance, but lowercase letters were given within the table. Finally, within the results, lines 165-166, a title was given for table 1 again, but it was given after table 2 and before table 3, thus, remove.

The phrase "without commitment to the horse's health or welfare" was deleted. Extra Title1 was removed too. Tables were re-arranged as requested by the journal. The use of SD was clarified at the bottom of the table.

As for the discussion, in line 181 add in brackets for references 9 and 10. Lines 193-200 repeat the results without giving additional interpretation. Either combine the results with the discussion or introduce within the discussion further interpretation of the data as it relates to other sports and as it relates to the welfare of these animals. The discussion section lacks relevancy to the readers and to a wider audience within the equine industry. While comparison with the endurance industry is given, it is limited in the discussion and does not address further studies reporting sports that may have fatalities, particularly those where speed is a component of the sport. It also lacks interpretation as it relates to utilizing this data for future studies. Lines 210-211 mention "the highly demanding degree that this sport represents for the horses", but what aspect of the sport are you referring to, ie. the speed, environment, etc? If speed is the "demanding degree", for example, how does that relate to other studies evaluating racehorses and fatalities and/or non-completion? 

The discussion spent much of the time repeating the results without further interpretation and comparisons to other research. The discussion should also include addressing study limitations. Being that this appears to be data taken after the fact, then, variability on what was documented, who documented it, and what information was given and recorded can impact the results. Also, since the study was over an expansive period, while it increased numbers, over the years methods within the race or within recording data could change impacting potential results more so than what was actually tracked and reported. Finally, no pre- collection was done to determine prior health nor were the animals tracked for an exceptionally long period to determine impact. In fact, it wasn't very clear how long after the race animals were tracked for those fatalities reported concerning after race deaths. Without a controlled study it is hard to contain potential variability that needs to be accounted for. Nevertheless, the study can still have merit concerning impact of the industry, even outside of this specific type of race, and impact for future studies, and thus, the authors need to clearly state these impacts within the discussion and summarize within the conclusion statement.

“Therefore, this study highlights the importance of characterizing RHU rides according to distance, evidencing that although LR are the most frequent, the proportion of deaths is higher in SR, which also determines a different profile in the causes of death and its risk factors.” was deleted

“From another point of view, completing the race was also a risk factor for fatalities. Not completing the race implied a higher percentage of deaths in the SR (96.6%) than in the LR (91%), although the SR had a greater number of horses that completed the race.” was deleted 

The phrase “the highly demanding degree that this sport represents for the horses” was changed by “The higher fatality rate of the RHU compared to the endurance competition, once again highlights the high metabolic and locomotor demand that this sport represents for the horse.”

First conclusion was deleted

A study limitation paragraph was added up in the discussion.

Some of the modifications maiden in order to improve relevance of the manuscript

Reviewer 3 Report

With interest, I have been reading the manuscript entitled: "Characterization of Raid Hipico Uruguayo Competencies by Ride Type (2007-2018): Causes of Death and Risk Factors " by Brito et al.

Congratulations on the manuscript. I have imagined how much hard work was involved, and it may have contributed to other researchers' training within the research group. The paper does not conform to the journal style. Of particular note, the authors have correctly used numbered citations but have yet to place these in the appropriate position in several phrases. Inside the tables is possible to see different types of formatting. I am not a native speaker, and while the manuscript seems well written in most parts, I think there are some mistakes and revisions by a native speaker that could improve the manuscript. The manuscript needs deep checking. Still, it presents some limitations. The following comments should be addressed before they are considered for publication.

Suggestions for improving the manuscript:

Simple summary

L-18: RHU has been cited for the first time. 

My comments: Please clarify it (Raid Hipico Uruguayo) 

Abstract:

My comments: Overall, the abstract onset does not address the research question adequately. Could you try to improve it?

Introduction

My comments: Well-written and straightforward. 

Materials and Methods

My comments: The MM needs to be more challenging to follow. I suggest authors present a study workflow (schematic representation of the experimental protocol).

Do you have any information on horses breed? Please, clarify it a bite more.

L:102-114: I needed clarification as the "Confort index (CI)" was used in the current study. The main problem with CI is that it can be misleading for many combinations of temperature and humidity. Reference number eight [8] (This reference needs to be updated) weakly supports this analysis. The validity of the "Confort index" as a variable categorization must be discussed. What low, medium, and high does mean? Was CI calculated by adding shade temperature? Please clarify it more, mainly in the discussion, as a study limitation. 

L106: "Causes of death were classified in four categories: metabolic conditions (colic, exhausted equine syndrome, disseminated intravascular coagulation)." 

My comments: How a disseminated intravascular coagulation diagnosis was performed? Please, clarify it a bite more.

Statistical analysis

Results and Discussion

Table 3: "Multivariate mixer logistic regression model"…

My comments: This reviewer is not convinced that this table helps much. It would be helpful for the reader to elaborate a bit further on how to interpret the multivariate mixer logistic regression (MMLR) results. Besides, the approach using a Multivariate mixer logistic regression model is exciting. However, my knowledge of this method is not sufficient to assess whether this method is appropriate in the context of evaluating causes of death and risk factors. This concept should be included when discussing and reviewing the results of the current study. Do you have any reference where MMLR was used for a similar purpose?

Conclusions

L249-250: This is not a conclusion. Do not have a link with the title or the study goal.

Author Response

We acknowledge the reviewer’s criticism and we state all corrections and comments below. We are now sending the revised version that we hope fulfill the requirements of ANIMALS. In order to be clearer, we have left all correspondence during this revision, and in italics letters our response.

The only changes we have made to the manuscript are at the request of the reviewers, but with 4 reviewers, there are many changes throughout the entire document.

We sent the manuscript to be reviewed by a native English speaker. We have not yet received the return of these corrections, the manuscript will be ready as soon as possible.

With interest, I have been reading the manuscript entitled: "Characterization of Raid Hipico Uruguayo Competencies by Ride Type (2007-2018): Causes of Death and Risk Factors " by Brito et al. Congratulations on the manuscript. I have imagined how much hard work was involved, and it may have contributed to other researchers' training within the research group.

Thank you very much for your words

 The paper does not conform to the journal style. Of particular note, the authors have correctly used numbered citations but have yet to place these in the appropriate position in several phrases. Inside the tables is possible to see different types of formatting. 

Thanks for the correction. The error was fixed

I am not a native speaker, and while the manuscript seems well written in most parts, I think there are some mistakes and revisions by a native speaker that could improve the manuscript. The manuscript needs deep checking. Still, it presents some limitations. The following comments should be addressed before they are considered for publication.

We sent the manuscript to be reviewed by a native English speaker. We have not yet received the return of these corrections, the manuscript will be ready as soon as possible.

Suggestions for improving the manuscript:

Simple summary L-18: RHU has been cited for the first time.  My comments: Please clarify it (Raid Hipico Uruguayo) 

Thanks for the correction. The error was fixed

Abstract: My comments: Overall, the abstract onset does not address the research question adequately. Could you try to improve it?

We added the next sentence into the abstract in order

We hypothesize that although this sport has similarities with other endurance disciplines, it has very different characteristics that affect the causes of death during races.

Introduction: My comments: Well-written and straightforward. 

Thank you very much for your words

Materials and Methods: My comments: The MM needs to be more challenging to follow. I suggest authors present a study workflow (schematic representation of the experimental protocol).

We did not fully understand the reviewer's requirement at this point. M&Ms were modified and some clarifications made according to other requests that we hope complete the requirements of the reviewer.

 Do you have any information on horses breed? Please, clarify it a bite more.

The breed of the horses competing in RHU was not officially registered until 2019. In an unpublished study (Brito et al.) of 305 horses that raced in one season of RHU, 237 were crossbreeds, 39 thoroughbreds, 15 Anglo-Arabians, and 14 Arabians horses. Horses called crossbreeds had mostly more than 75% thoroughbred blood.

This info was added to the intro

L:102-114: I needed clarification as the "Confort index (CI)" was used in the current study. The main problem with CI is that it can be misleading for many combinations of temperature and humidity. Reference number eight [8] (This reference needs to be updated) weakly supports this analysis. The validity of the "Confort index" as a variable categorization must be discussed. What low, medium, and high does mean? Was CI calculated by adding shade temperature? Please clarify it more, mainly in the discussion, as a study limitation. 

The comfort index was taken from Jones 2008. Although the denomination “comfort index” is confusing, Probably discomfort index is more appropriate, but refers to another index. So, we kept the original name (CI) to avoid adding confusion. 

High, medium and low represent the level of discomfort. The classification, proposed by Jones, also simplifies the inclusion of the model (trichotomization).

It is currently used by the FHU, due to its simplicity. Many indices have been evaluated to assess heat stress, even in horses, showing a better predictive capacity. We added a sentence in the discussion in reference to this, according to this observation.

The comfort index was used as an indicator of thermal stress in this work. It is widely used due to its simplicity and low cost, but there are significant weaknesses due to misleading for many combinations of temperature and humidity.  Many indices have been evaluated to assess heat stress, even in horses, showing a better predictive capacity.This limitation should be considered for the interpretation of the results.

L106: "Causes of death were classified in four categories: metabolic conditions (colic, exhausted equine syndrome, disseminated intravascular coagulation)." My comments: How a disseminated intravascular coagulation diagnosis was performed? Please, clarify it a bite more.

the data is from the autopsy report. We don't have access to more than that. The findings are petechial hemorrhages and peripheral or mucosal ecchymoses, and visceral enlargement, mainly in the liver and kidneys. Hemorrhages also appear. It was described in endurance (Muñoz et al, Current knowledge…). The number of animals with ICS that we found is striking.

Statistical analysis

Results and Discussion

Table 3: "Multivariate mixed logistic regression model"…

My comments: This reviewer is not convinced that this table helps much. It would be helpful for the reader to elaborate a bit further on how to interpret the multivariate mixer logistic regression (MMLR) results. Besides, the approach using a Multivariate mixer logistic regression model is exciting. However, my knowledge of this method is not sufficient to assess whether this method is appropriate in the context of evaluating causes of death and risk factors. This concept should be included when discussing and reviewing the results of the current study. Do you have any reference where MMLR was used for a similar purpose?

Thank to reviewer for your words about the model (MMLR). We are very pleased that you enjoyed that we applied MMLR. We believe that the data benefit from being presented in the Table 3, since the results allow a clear visualization of the interpretation of the variable in the models. Other authors (Fielding et al., 2011; Legg et al., 2019; Nagy et al., 2010, 2014) with similar studies use similar presentations, which help the interpretation of our results by the readers. The risk studies should be presented together with the results indicating the factors involved in the model (N Ride, Observations). The table is self-explanatory, as it provides information regarding the variables (Short and Long race and their Predictors) and the benefit of including the random factor in the mixed model (Marginal R2/Conditional R2). If the predicted values for the residuals include the random effects, they are called Conditional R2 statistics, as opposed to if the random effects are excluded from the calculation of the predicted values leading to the residuals they are called the Marginal R2 statistics. Berridge & Crouchley (2011) in book “Multivariate Generalized Linear Mixed Models Using R” presents similars analysis methodology that we use in us data. Our work is original in view that the outcomes are modeled as a predictor variables when data are both fixed and random effects. Considering the sample size of the studied and the beneficit of the the interpretation of the results with the variability related to the Ride. 

Conclusions L249-250: This is not a conclusion. Do not have a link with the title or the study goal

Thanks for the correction. The paragraph was moved to discussion section

Reviewer 4 Report

Dear Authors,

This is a very interesting paper concerning the causes of injuries in endurance competition. The more we know about the adaptations to training and exercise, the better the animal can compete, maintaining its welfare and allowing express its excellent fitness and level of training. I really enjoyed reading this. In addition, the importance od the study is high. The manuscript entitled “Characterization of Raid Hipico Uruguayo competencies by ride type (2007-2018): causes of death and risk factors " has an appropriate title. However, it should be rewritten in some parts for better understanding so some corrections are needed.

Please find some specific comments:

Simple summary:

L25 – change metabolic death to metabolic alternations

Introduction:

L80: In my opinion the paragraph about parameters which may help in monitoring horses adaptation to increasing workload during training is important. There are some studies concerning an influence of regular training in horses which should be considered. There are the physical activity-dependent hematological and biochemical changes in endurance horses as well as changes in Serum Amyloid A (SAA). During the training not only acute phase protein production (such as SAA) is changing but also the cytokine production leading to creation the anti-inflammatory state in horses introduced to the endurance training. In addition, there are some novel cytokines such as IL-13 which production strongly influences on metabolic conditioning of muscle to endurance exercise in humans and in horses. Also, it was suggested the ROS production is different in horses at different fitness level. It was published that, β2-Adrenergic stimulation is strongly influencing on ROS production and depends on fitness level in racehorses. Adding this information may be beneficial. However, one ideal marker does not exist. Thus such information should be included.

Discussion:

I will change the ride- type - short to medium because long would be rather 140-160km.

Also the part about injuries prevention should be added.

Author Response

We acknowledge the reviewer’s criticism and we state all corrections and comments below. We are now sending the revised version that we hope fulfill the requirements of ANIMALS. In order to be clearer, we have left all correspondence during this revision, and in italics letters our response.

The only changes we have made to the manuscript are at the request of the reviewers, but with 4 reviewers, there are many changes throughout the entire document.

We sent the manuscript to be reviewed by a native English speaker. We have not yet received the return of these corrections, the manuscript will be ready as soon as possible.

Dear Authors, This is a very interesting paper concerning the causes of injuries in endurance competition. The more we know about the adaptations to training and exercise, the better the animal can compete, maintaining its welfare and allowing express its excellent fitness and level of training. I really enjoyed reading this. In addition, the importance of the study is high. The manuscript entitled “Characterization of Raid Hipico Uruguayo competencies by ride type (2007-2018): causes of death and risk factors " has an appropriate title. However, it should be rewritten in some parts for better understanding so some corrections are needed.

Thank you very much for your inspiring words. We are very pleased that you enjoyed the manuscript.

Please find some specific comments:

Simple summary: L25 – change metabolic death to metabolic alternations

Thanks for the correction. metabolic death was changed by death due to metabolic alterations thorough the entire manuscript.

Introduction: L80: In my opinion the paragraph about parameters which may help in monitoring horses adaptation to increasing workload during training is important. There are some studies concerning an influence of regular training in horses which should be considered. There are the physical activity-dependent hematological and biochemical changes in endurance horses as well as changes in Serum Amyloid A (SAA). During the training not only acute phase protein production (such as SAA) is changing but also the cytokine production leading to creation the anti-inflammatory state in horses introduced to the endurance training. In addition, there are some novel cytokines such as IL-13 which production strongly influences on metabolic conditioning of muscle to endurance exercise in humans and in horses. Also, it was suggested the ROS production is different in horses at different fitness level. It was published that, β2-Adrenergic stimulation is strongly influencing on ROS production and depends on fitness level in racehorses. Adding this information may be beneficial. However, one ideal marker does not exist. Thus such information should be included.

It would be very interesting to be able to incorporate training concepts and biochemical markers, and compare them with fatalities. Unfortunately this is a retrospective study and we do not have much more data than the one exposed, in fact the identity of all the horses and owners is unknown. We are currently designing a prospective trial (with elimination, not deaths) and these concepts will definitely be taken into account.

Discussion: I will change the ride- type - short to medium because long would be rather 140-160km.

We consider it appropriate to use the ride type (long and short) proposed by FEU. The sport has commonalities with FEI endurance, but it is different in many ways, and we did not want to confuse the readers with an idea of moderate distance FEI endurance. We include the following sentence in the discussion, and if the reviewer agrees, we will keep the denomination long and short

The racing distances were classified as short and long according to the FHU regulations. Long FEI endurance races would be longer than 140 km. 

Also the part about injuries prevention should be added.

Many variables and variability associated with horse background and the environment could not be recorded, so this info has to be interpreted with caution, and is not easy to make practical recommendations. The following sentence was exposed at the end of the discussion

further studies are needed to assess more specific risk factors, in order to provide objective data that can help planning racing schedules and serve as a basis for regulations, ultimately improving both the welfare of RHU horses and the public perception of this discipline

Round 2

Reviewer 3 Report

The reviewer would like to thank the authors for taking the time to provide additional analysis and discussion of their research data and for making extensive additions to the manuscript to address comments from the initial review. The authors have addressed all recommendations for revision, and therefore the reviewer recommends accepting the manuscript for publication after English checking.

Author Response

Thank you very much for reviewing the manuscript. your work has improved the manuscript, and has left deep knowledge in the work team that will be useful in our future as researchers.

The English review will be ready for Tuesday. If you want to see the final version, it will be in the system on Tuesday at noon Argentina time.

Kind regards and thanks

Reviewer 4 Report

I will still add some information about possibilities connected with the injuries prevention. However, I leave the decision to the Editor.

Author Response

The authors regret that we were unable to meet the reviewer's requirements. However, we thank you very much for reviewing the manuscript. your work has improved the manuscript, and has left deep knowledge in the work team that will be useful in our future as researchers.

The English review will be ready for Tuesday. If you want to see the final version, it will be in the system on Tuesday at noon Argentina time.

Kind regards and thanks